

# Assessment of avalanche hazard of freeride skiing areas

Adam Kupec[1], Štefan Koco[1, 2]

[1]University of Presov in Presov, Faculty of Humanities and Natural Sciences, Department of Geography, Presov, 080 01, Slovakia

[2]National Agricultural and Food Centre – Soil Science and Conservation Research Institute, Presov, 080 01, Slovakia

*Correspondence to*: Adam Kupec (adam.kupec@smail.unipo.sk), Štefan Koco (stefan.koco@unipo.sk)

**Abstract.** Freeride skiing and ski touring have been growing in popularity in Slovakia as an alternative to crowded ski slopes, offering skiers the thrill of untracked snow and challenging terrain. However, venturing into unmanaged mountain areas exposes participants to significantly greater dangers, especially avalanches and risk of falling. This study presents an approach

for assessing avalanche hazard of freeride areas, demonstrated at the Jasná ski resort in Slovakia's Low Tatras. Using high-quality elevation data, precise vegetation mapping, and historical avalanche records, potential avalanche release zones were identified, their potential run-out paths for skier trigger avalanches (≤ size 3) were modelled, and the frequency in which are avalanches likely to occur on different slopes were approximated. Results show that 15.9% of the area has high to very high release potential, with the most hazardous slopes concentrated on steep, north-facing terrain above 1700 metres. Simulations

of more than 180 avalanche scenarios produced run-outs covering 44% of the area. Frequency analysis found that 64.9% of avalanche-prone slopes in freeride zones are subject to very frequent activity. Moreover, frequency approximation achieved 82,61% match with intersecting areas of the existing avalanche cadastre. Based on the results the freeride zones were divided into 4 groups based on their danger level. The proposed approach can be adapted to other mountain regions and may be further improved by automating vegetation mapping, modelling additional avalanche types, and using open-source simulation tools.


## 1 Introduction

In recent years, a notable increase in interest in adrenaline sports has been observed. This interest is often manifested particularly in sports that have a long tradition. In Slovakia, such sports include skiing, as the mountain ranges like High Tatras or Low Tatras provide suitable conditions for the development of world-class ski resorts, although these conditions are

changing due to global warming. Nevertheless, some skiing enthusiasts find overcrowded slopes of ski resorts less appealing and choose to explore unmaintained terrains through freeride skiing. While thrilling, freeride skiing presents a variety of new challenges and risks compared to conventional skiing. In addition to navigating steeper slopes and untracked snow, participants must be aware of potential hazards such as protruding rocks and, most importantly, the threat of avalanches. Unfortunately, many individuals lack the necessary training to accurately assess the safety of various slopes, often leading to a dangerous

underestimation of avalanche risks. This lack of awareness can result in serious injuries or tragic fatalities. Therefore, the mitigation of the hazard through localization of hazardous areas and creation of easily interpretable representations should be our main focus. Although the majority of mountain ranges in Slovakia are mapped in terms of avalanche run-outs, the accessibility of this data is limited, and in some instances, the data are outdated.



The systematic study and mitigation of avalanche impacts in Slovakia began in 1924, after the occurrence of the deadliest
avalanche recorded in Slovak history, which resulted in the fatalities of 18 individuals in the settlement of Rybô (SLP HZS
2024). Since then, a dedicated avalanche research facility (Slovak Centre for Avalanche Prevention) was established as a
specialized research institution dedicated to the investigation of snow dynamics, the implementation of avalanche control
measures, the prediction of avalanche hazards, and the advancement of the avalanche work as a science.

A significant contribution of the Slovak Centre for Avalanche Prevention (SCAP) is the Avalanche Cadastre, developed
by Milan in 1981, which documents nearly four thousand distinct avalanche paths. However, given the nearly five-decade age
of this Cadastre, it is increasingly evident that some of the documented avalanche run-outs may no longer exist or that their
trajectories have shifted due to changes in land cover. Consequently, an update of this Cadastre is required. Nevertheless, the
Cadastre remains a benchmark for evaluating the accuracy of avalanche modelling.

Recent technological advancements in Geographic Information Systems (GIS) and remote sensing have enhanced the
capacity to estimate avalanche trigger zones and run-out areas without prolonged observational studies. Over the past two
decades, a plethora of scientific investigations was conducted by both Slovak and international researchers to estimate probable
avalanche trigger zones (Hreško 1998; Hreško and Bugár 2000; Barka 2003; Rybár and Barka 2003; Sykes, Haegeli, Bühler
2022), model avalanche run-outs and assess avalanche hazard (Biskupič and Barka 2009; Biskupič and Richnavský 2010;
Boltižiar, Biskupič and Barka 2016; Oller, Baeza and Furdada 2021; Košová, Molokáč, Čech 2022; Ortner et al. 2023).
Furthermore, studies have aimed to reconstruct historical avalanche events (Biskupič et al. 2011, Aydin et al. 2014), critically
evaluate the accuracy of different avalanche simulation models (Christen, Bartelt, Kowalski 2010a; Martini, Baggio,
D'Agostino 2023) and develop automated standalone tools/models for large-scale avalanche terrain classifications (Bühler et
al. 2022, Toft et al. 2024, Statham and Campell 2025). These studies have predominantly employed well-established models
like RAMMS: Avalanche (Christen et al. 2010b), SAMOS-AT (Sampl and Zwinger 2004) or Alpha-Beta regression model
(Lied and Bakkehoi 1980). However, these works rarely focus on freeride or backcountry skiers (Schmudlach and Köhler
2016, Harvey et al. 2018) and identification of avalanche-prone zones in freeride areas, which are arguably the most dangerous.

The objective of this study is to utilize well-established tools to develop an avalanche terrain map specifically for skier-
triggered avalanches (≤ size 3, as classified by European Avalanche Warning Services) within freeride ski areas. This will be
accomplished by combining traditional numerical methods utilized in avalanche science with the RAMMS model and very
high-resolution Digital Elevation Model (DEM) data (1m per pixel) and recent (2022) vegetation data. Additionally, the
findings of the proposed approach are intended to be accessible and interpretable by non-professional audience. With minor
adaptations, such as reclassifying input factors to more accurately represent the characteristics of particular areas, the approach
should be applicable to various geographical regions to identify slopes susceptible to avalanche hazards. We believe that
achieving this goal could enhance the awareness and increase the safety of freeride skiers and mountaineers who may be
exposed to avalanche risks while engaging in their favourite activities.

**2 Study area**





This study examines the region surrounding the Jasná ski resort, situated within the Low Tatras, the second-highest mountain range in Slovakia. Renowned for its exceptional skiing experience, the Jasná ski resort features numerous freeride areas in its vicinity, which attract both freeride enthusiasts and mountaineers. The designated freeride areas cover approximately 7.67 km² and are subdivided into twelve distinct zones. They are characterized by a variety of steep, unforested slopes with diverse aspects that exhibit high levels of avalanche activity (Jasná 2017).

**Figure 1.** Overview of study area. Red rectangle within the map of Slovakia represents the location of study area. Backgorund DEM: ÚGKK SR, 2024. Freeride areas localization and difficulty: Jasná Freeride manual, 2017.

The northern slopes of the mountain range are distinguished by their rocky terrain, exhibiting steeper gradients (ranging from 30 to 50 degrees) and numerous gullies. In contrast, the southern slopes are predominantly meadows with a less steep



gradient (less than 35 degrees). The elevation of the freeride zones on both sides of the range varies from approximately 1350 meters above sea level (m a.s.l.) to 2024 m a.s.l. Despite the differences in topography, both slope types provide suitable conditions for the formation of avalanche release zones.

In addition, freeride zones are, in the freeride manual (Jasná 2017), systematically categorized into three levels of difficulty. Five zones are classified as easy, with slope inclinations ranging from 25 to 30 degrees, six zones are graded with medium difficulty, characterized by slope inclines ranging from 31 to 36 degrees, lastly, only a single zone is classified as difficult, with slope inclines that range from 37 to 50 degrees. The location of study area is depicted in Fig. 1.

The skiing season usually begins in early December and concludes at the end of April, with the avalanche danger persisting throughout the entire period. Annually, several incidents involving individuals being caught or even killed due to avalanches occur in this region.

## 3 Methods

### 3.1 Data sources

While avalanche modelling has the potential to be highly representative of real-world phenomena, its accuracy is fundamentally influenced by the quality of the input data, particularly when simulating smaller-scale avalanches (Bühler et al., 2011; Miller et al., 2022). The basis for our work was a Digital Elevation Model (DEM) in raster format, with a resolution of 1 m²/pixel. The DEM provided the essential basis for all topographic factors incorporated into the avalanche release zone estimation model established by Biskupič and Barka (2009), as well as for subsequent simulations performed using the RAMMS framework.

To account for the roughness input factor, a land cover layer had to be developed, as for the territory of the Slovak Republic, high-quality data are not available. This layer was constructed by mapping land cover from aerial imagery, with a resolution of 20cm per pixel. Both, the DEM and the aerial imagery were obtained from the official website of the Department of Geodesy, Cartography, and Cadastre of the Slovak Republic (www.geoportal.sk). Furthermore, avalanche records collected from the SCAP, were crucial for calibration of input factors utilized within the Biskupič and Barka model (2009).

### 3.2 Statistical analysis of SCAP avalanche records

The SCAP systematically documents all reported avalanche incidents within the avalanche-prone mountain ranges of Slovakia. The data collected by SCAP serves as a vital resource, encompassing various parameters related to avalanche dynamics. Specifically, it provides detailed information regarding release zones, including attributes such as elevation, geomorphological characteristics, snow type, aspect, and causative factors. Additionally, the dataset covers transport zones, describing aspects such as morphology, length, width, and movement patterns, alongside deposition zones which contain the information about the height and dimensions of deposited materials.

Moreover, the records include statistics about the number of individuals buried by snow, number of casualties, extent of infrastructural damage and predicted avalanche danger levels for a specific day.

For the accurate calibration of the avalanche release zone estimation model developed by Biskupič and Barka (2009), the SCAP data provided essential insights, particularly regarding the exposition and altitude of release zones. The analysis of




avalanche records enabled the adjustment and refinement of the model to better suit the specific conditions of our area of
interest.

**3.3 Avalanche trigger zones estimation**

As previously mentioned, the estimation of avalanche trigger zones utilizes the mathematical model proposed by
Biskupič and Barka (2009). This model represents the most recent advancement in potential release areas delineation within
the mountain ranges of the Slovak Republic. Initially introduced by Hreško (1998), the model underwent modifications by
Rybár and Barka (2003) before being refined to its final iteration by Biskupič and Barka.

The mathematical formulation of this model is characterized by several key factors: Av represents the probability of
avalanche release (with a higher value indicating a greater likelihood), Al represents the altitude factor, Ex refers to the
exposure/aspect factor, Fx refers to the profile curvature factor, Fy represents the plan curvature factor, S denotes the
slope/inclination factor, and Rg refers to the roughness factor, encompassing vegetation cover.

The Biskupič and Barka model (2009) operates on the principle of assigning numerical values ranging from 0 to 2 to
each component (in the case of the land cover factor from 0 to 3) of the factor, based on the extent of impact each component
has on the potential avalanche trigger (where a higher value corresponds to a stronger impact). Numerical values for the
exposure (Ex) and altitude (Al) factors were derived from a statistical analysis of SCAP's avalanche records, while the values
for the other factors remained the same with those established in the original work of Biskupič and Barka (2009), given that
their impacts are uniform across all mountain ranges in the Slovak Republic. Before assigning values presented in Tab. 1, focal
analysis (average on the 5x5 cells area) of raster containing information about slope angles was conducted to eliminate the
impact of individual cell errors on final classification.

**Table 1.** Biskupič and Barka (2009) avalanche trigger estimation model formula, values attributed to each factor adjusted for
study area and results reclassification

$$Av = (Al + Ex + Fx + Fy) \times S \times Rg$$

| Altitude (m a.s.l.) | Altitude factor (Al) | Profile curvature (%) | Profile curvature factor (Fx) | Plan curvature (%) | Plan curvature factor (Fy) |
|---|---|---|---|---|---|
| 1250 - 1450 | 0,1 | <4 – 0,2) | 1 | <-4 – -0,2) | 1 |
| 1450 - 1650 | 0,5 | <0,2 – -0,2) | 1 | <-0,2 – 0,2) | 1 |
| 1650 – 1850 | 2 | <-0,2 – -0,5) | 1 | <0,2 – 0,5) | 1 |
| > 1850 | 1,5 | <-0,5 – -4> | 0,5 | <0,5 – 4> | 0,5 |

| Land cover type | Roughness factor (Rg) |
|---|---|
| Build-up area | 0 |
| Forest (coniferous, deciduous, mixed) | 0,5 |
| Open forest with dwarf-pine, rough stone debris and slopes covered by smaller blocks | 1,2 |
| Shrub vegetation | 1,4 |





| Open forest | | 1,5 | |
| Compact dwarf-pine and slopes with rock outcrops up to 50cm | | 2,5 | |
| Grass with sporadic dwarf-pine and fine slope debris | | 2,8 | |
| Grass areas, blockfields and rock plates | | 3 | |
| **Slope (°)** | **Slope factor (S)** | **Exposition** | **Exposition factor (Ex)** |
| | | N | 1,7 |
| <0 – 10); <70 – 90> | 0 | NE | 2 |
| <10 – 19); <60 – 70) | 0,4 | E | 1 |
| <19 – 25); <55 – 60) | 0,8 | SE | 0,8 |
| <25 – 30); <50 – 55) | 1,2 | S | 0,7 |
| <30 – 35); <45 – 50) | 1,6 | SW | 0,4 |
| <35 – 45) | 2 | W | 0,5 |
| | | NW | 1,5 |
| **Equation result (Av)** | | **Avalanche release probability** | |
| 0 - 15 | | Low | |
| 15 – 22.5 | | Medium | |
| 22.5 - 30 | | High | |
| 30 - 36 | | Very high | |

Upon assigning values to each factor, calculations were executed within QGIS. Values of the Av factor ranged from 0 to 36 (the higher the number, the greater the avalanche release potential of a specific area). Subsequently, the Av layer was reclassified into four distinct categories based on the probability of avalanche release (as detailed in Table 1). Potential avalanche trigger zones, used as starting points of simulations, were then mapped out from the layer, where the values of Av factor cells were ≥ 15 and cells created a cohesive unit.

**3.4 Avalanche run-out modeling**

Avalanche run-out zones were calculated in the RAMMS simulation model. RAMMS is a two-dimensional numerical simulation framework developed by the WSL Institute for Snow and Avalanche Research SLF in Switzerland. It serves as an alternative to conventional one-dimensional avalanche simulation tools such as AVAL-1D. The model is based on the Voellmy-Salm approach, enabling the computation of accurate avalanche run-out distances, flow velocities and impact pressures (Christen et al. 2010).

For the simulation of run-outs of skier-triggered avalanches, variable friction parameters (dry-Coulomb type friction (coefficient $\mu$) and viscous-turbulent friction (coefficient $\xi$)) were, based on friction calibration tables, set to "Tiny" (coefficient $\mu$ ranged from 0.275 – 0.47 and coefficient $\xi$ ranged from 900 to 1750 depending on the type of terrain) with a return period of 10 years. Altitude limits were slightly adjusted to better represent the studied region. The cohesion parameter was set to 50 as recommended by the RAMMS User Manual (SLF 2024). The density of the snow was kept at 300kg/m³, and the release depth of all simulated avalanches was set to 50 cm, reflecting the mean fracture depth of human-triggered avalanches in



Switzerland (Schweizer and Lütschg 2001). The resolution of the Digital Elevation Model (DEM) was resampled from 1m per pixel to 2m per pixel to reduce computation time without compromising the quality of the results and to achieve smoothing

effect, so the DEM resembles winter terrain better (Dreier et al. 2014, Miller et al. 2022). The forest layer, essential for the accurate simulation of avalanches that interact with the tree line, was obtained from the vegetation cover layer created for avalanche trigger zones estimation.

**3.5 Frequency estimation**

Frequency estimation was conducted through a synthesis of the results of avalanche trigger estimation model of Biskupič

and Barka (2009) and the outcomes of run-out calculations performed in RAMMS. In our understanding, an increased probability of avalanche triggering correlates with a higher frequency of avalanche occurrences. Therefore, avalanche run-outs originating from regions identified, according to the Biskupič and Barka (2009) model, as having a predominantly very high or high probability of triggering (≥50% of the area) were classified as slopes with a very frequent occurrence of avalanches. Slopes characterized by trigger zones predominantly comprised of moderate probability cells, with a lower percentage of high

and very high probability cells (≥25% of the area, but ≤50% of the area), were categorized as slopes with a frequent occurrence of avalanches. Lastly, avalanche run-outs whose initiation zones were primarily composed of cells with moderate probability (≥50% of the area), lacking or featuring a low number of high probability cells (≤25% of the area), were designated as exhibiting sporadic avalanche occurrences.

**3.6 Overall freeride zone hazard assessment**

The final step involved conducting a comprehensive hazard assessment of freeride zones, which was based on avalanche hazard mapping. The freeride zones were categorized into four distinct groups: safe, moderately dangerous, dangerous and very dangerous.

Freeride zones were considered safe if avalanche slopes or extreme terrain were not present. Zones were classified as moderately dangerous if they primarily featured slopes that were susceptible to sporadic avalanches. Those categorized as

dangerous were characterized by a predominance of slopes with frequent avalanche occurrences, while zones classified as very dangerous exhibited slopes prone to very frequent avalanches.

In addition to this classification, the potential risk of falling due to extreme terrain (>50°) was considered and incorporated as an auxiliary factor in the hazard evaluation.

**4 Results**

**4.1 Statistical analysis of SCAP records**

The SCAP database consisting only of avalanche records relevant to the case study area contained a total of 207 avalanches occurring within the span from 2004 to 2023. For our analysis all avalanches were incorporated, encompassing both human-triggered and naturally occurring events, classified up to size category 3. The annual frequency of reported

incidents demonstrates variability, with a notable decline in recent years. This reduction is attributable to the SCAP's methodological shift, implemented in 2014, which refocused from the documentation of all avalanches to the recording of avalanche accidents. Nevertheless, certain avalanches, primarily reported by skiers or ski patrols, continue to appear within





the database. It is important to note that the quality of reports varies significantly; some are thoroughly detailed, while others are minimally documented.

190        As mentioned previously, the statistical analysis primarily focused on the aspect and elevation of the avalanche release areas. The aspect was recorded in 206 instances, typically categorized according to the eight cardinal directions (N, NE, E, etc.). In certain cases, data was also recorded in secondary intercardinal directions (NNE, ENE, ESE, etc.). For these instances, reports were adjusted to align with the nearest cardinal direction.

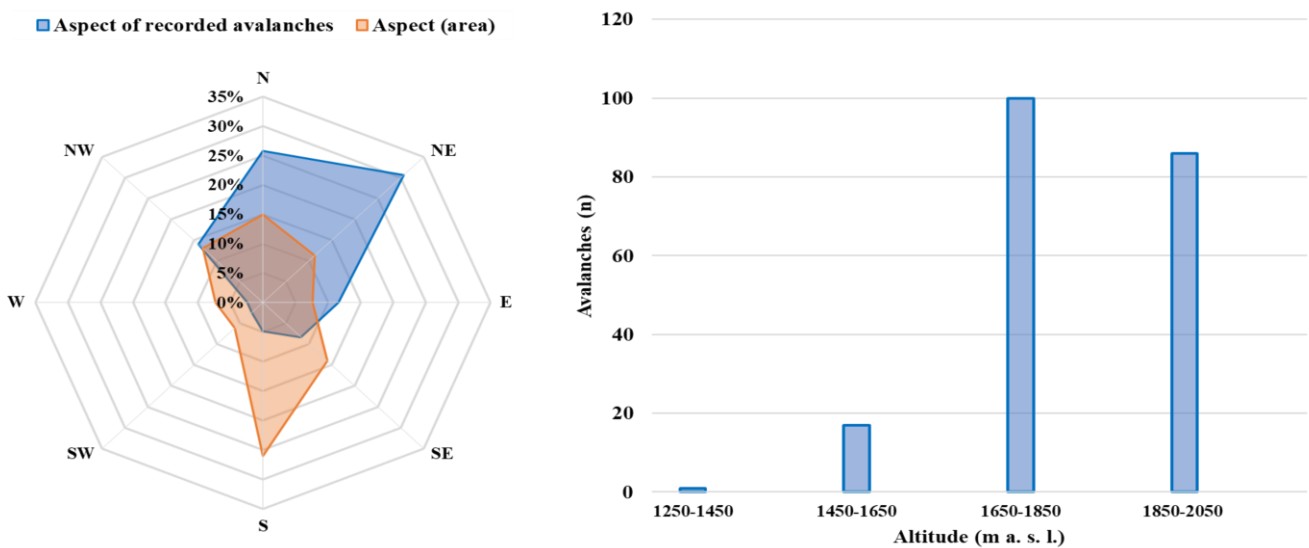

**Figure 2.** Results of aspect and altitude analysis of avalanches recorded in study area.

       The analysis revealed that the predominant aspect of these avalanches was northeast, accounting for 31%, followed closely by north at 26% and northwest at 14%. In contrast, southern aspects (S, SE, and SW) exhibited lower avalanche activity, collectively representing only 16% of the recorded events. The remaining avalanches were triggered in the eastern aspect 200  (12%) and the western aspect (2%).

       The avalanche records contained data about elevation in 204 cases. The statistical analysis indicated that most avalanches (100 – 49,06%) were triggered within the altitude range from 1650 to 1850m a.s.l. In the higher areas, above 1850m a.s.l., 86 avalanche releases were recorded and in the altitudes between 1450 - 1650m a.s.l., 17 avalanche triggers were found. Notably, only one avalanche trigger was registered in areas below 1450 m a.s.l. The median altitude of the starting zones in 205  the area is 1750m a.s.l, with a mean altitude of 1745m a.s.l. The visual representation of the analysis results is portrayed in Fig. 2.

## 4.2 Avalanche trigger zones estimation

       Results from the estimation model indicate a notable correlation with the avalanche cadastre map; however, the findings regarding trigger zone estimation encompass even the areas that are not included within the perimeter of the avalanche





cadastre. This discrepancy is particularly evident on southern-facing slopes. In contrast to the work presented by Biskupič and Barka (2009) forest and dwarf pine succession is not observable in higher parts of the study area, therefore, avalanche danger remained relatively unchanged. The model reveals that 64.14% of the study area exhibits a low probability of avalanche triggering, 20.14% demonstrates a moderate triggering probability, 10.80% is characterized by a high triggering probability, and 4.92% is classified as possessing a very high probability of avalanche initiation. Notably, the areas with the highest

potential for avalanche release are situated on northern slopes above 1700m a.s.l.

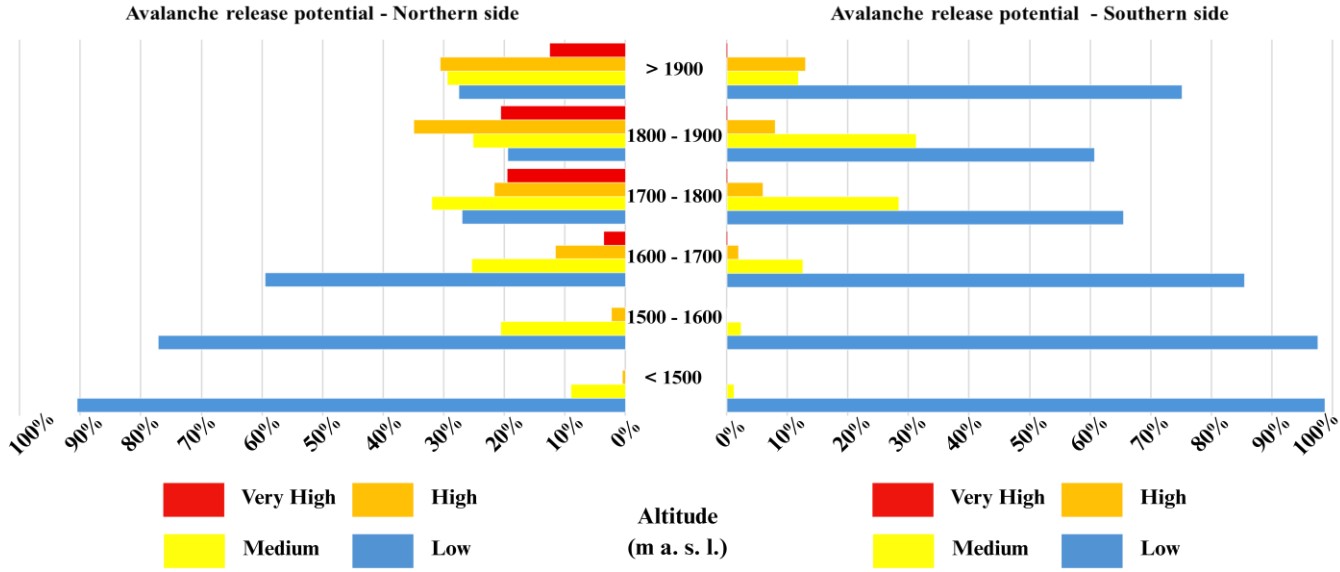

**Figure 3.** Avalanche release potential on the northern and southern sides of the study area in relation to altitude.

       As previously noted, the topography of the northern and southern slopes of the mountain range differs, which,

according to the results of avalanche trigger estimation, is also reflected in their trigger potential. Figure 3 illustrates the avalanche release potential on the northern and southern sides of the study area in relation to altitude. The findings indicate that the northern slopes exhibit a significantly greater percentage of cells classified with very high and high release probabilities (especially in elevations exceeding 1700 meters above sea level) compared to the southern slopes, where cells indicating very high probability are almost entirely absent.

**4.3 Avalanche run-out modelling**

       For avalanche run-out mapping, over 180 individual avalanche simulations were conducted. Simulated avalanche run-outs cover approximately 44% of the study area. Similar to the outcomes derived from the estimation model, the results of the run-out simulations demonstrate a notable correlation with the avalanche cadastre. However, it is important to note that the simulated run-outs are, in certain instances, shorter than the established avalanche paths documented in the SCAP cadastre.

This discrepancy is primarily attributable to the restriction of simulations to avalanches of size ≤ 3.



The degree of overlap between the simulated run-out areas and the avalanche cadastre is approximately 62,71%. The difference observed can be attributed to simulated run-outs originating from trigger zones that do not correspond with the avalanche cadastre, particularly on slopes with southern exposure. Without the aforementioned slopes, the overlap percentage increases significantly, reaching nearly 80%. Moreover, it can be assumed that accuracy of the model is even higher

considering the inaccuracies in the avalanche cadastre (Milan 1981).

**4.4 Avalanche frequency estimation**

The results of the avalanche frequency estimation, in which the avalanche trigger estimation model developed by Biskupič and Barka (2009) was utilized, indicate that slopes exhibiting sporadic avalanche occurrences account for 25,79% of avalanche-prone areas. Slopes characterized by frequent and very frequent avalanche activity cover 9,25% and 64,96% of the

avalanche run-out zones, respectively.

To the overall expanse of the freeride area, slopes with sporadic avalanche occurrences represent 11,48%, those with frequent avalanche activity cover 4,12%, and slopes demonstrating very high avalanche frequency make up 28,92% of the total freeride zones.

A comparative analysis of the frequency data recorded in the avalanche cadastre alongside the estimations derived

from the Biskupič and Barka model (2009) indicates a probability of detection of 82,61% on the intersecting areas.  It is important to note that in the majority of nonmatching areas, the presented approach tends to exaggerate the avalanche frequency, attributing higher frequency to regions that, based on the cadastre, have lower avalanche occurrence. Figure 4 portrays the extent of avalanche run-outs with frequency of their probable occurrence (simulated) overlapped with avalanche cadastre created by Milan (1981).





**Figure 4.** The extent of avalanche run-outs with estimated frequency of their probable occurrence overlapped with avalanche cadastre created by Milan (1981). Backgorund DEM: ÚGKK SR, 2024. Avalanche cadastre data: Milan, 1981.

## 5 Discussion

### 5.1 Overall freeride zone hazard assessment

For the final assessment, auxiliary factor of extreme terrain (>50°) was also considered and served as intensifier of overall hazard. Based on a figure 5, it can be concluded that one freeride zone (Zone 10) was classified as safe, as it lacks both avalanche-prone slopes and extreme terrain features. Two freeride zones (Zone 11 and 12) were classified as moderately dangerous. Their classification to said hazard level can be attributed to the sporadic occurrence of avalanches in these zones, moreover, extreme terrain is absent in both cases. Two freeride zones (Zone 4 and Zone 5) were categorized as dangerous. In



case of Zone 4, primarily due to the prevalence of slopes with frequent avalanche occurrences on most of their slopes, with a notable exception of one slope with very frequent avalanche activity. In case of the latter one, the classification as dangerous was based predominantly on the presence of extreme slopes. The remaining seven freeride zones are classified as very dangerous. All of the zones have slopes with very frequent avalanche activity, alongside the presence of extreme terrain. Visual representation of the classification is shown in Fig. 5.

**Figure 5.** Final classification of freeride zones based on their hazard level. Based on the classification, one freeride zone was classified as safe (blue), 2 as moderately dangerous (yellow), two as dangerous (orange) and 7 freeride zones were found to be very dangerous (red). Backgorund DEM: ÚGKK SR, 2024. Freeride areas localization: Jasná Freeride manual, 2017.

Correlation between the difficulty of the freeride zones and their overall hazard is observable. The higher the difficulty of a freeride zone, the greater the avalanche hazard. The correlation is based on the steepness, as difficulty in the freeride manual





was also set based on the inclination of slopes within freeride zones. While easy freeride zones were, based on their hazard, classified as safe, moderately dangerous or in two instances as dangerous, difficult and medium difficult freeride zones were all marked as very dangerous.

**5.2 Strengths of the approach and limitations**

The methodology proposed in this paper integrates the avalanche trigger estimation model proposed by Biskupič and Barka (2009) and run-out distances modelled in system RAMMS. The approach shows good applicability in the areas, where historical avalanche records are accessible for correct calibration of the model. Furthermore, the trigger estimation considers a wide range of factors, without the requirement of a multitude of input data. This makes it easily applicable in the areas where
Digital Elevation Models (DEM) or Digital Surface Models (DSM) and vegetation data are available. Additionally, the estimation model can also be used to determine the frequency of the avalanches on specific slopes. Ultimately, the proposed approach enables the development of an avalanche cadastre for any location susceptible to avalanche activity.

The main limitation of avalanche trigger model is that it heavily relies on the resolution of DEM. Although, model proved effective even when applied on the DEM with the resolution of 10m/pixel (Biskupič and Barka 2009), its applicability
may be harder in the areas where high-quality digital models are absent. Apart from reliance on the DEM resolution, model relies on up-to-date vegetation data, which can change relatively quickly. Moreover, vegetation data in Slovakia is available only in coarse resolutions (25m and lower) which may not be suitable if we want to simulate small, i.e. skier-triggered avalanches. Mapping vegetation manually is time consuming, which means applicability is time demanding, especially on larger areas. This inconvenience could be overcome by implementing Semi-automated classification or by subtraction of DEM
from DSM (Brožová et al. 2020, Brožová et al. 2021). Moreover, applicability may be hindered if we try to apply the model on the areas with no avalanche records, as calibration becomes impossible, the closest way is to find area with similar features and apply them for the area. In case of unknown area, as well as large-scale application, other avalanche trigger estimation models or avalanche terrain classification models should be considered (Bühler et al. 2013, Bühler et al. 2018, Bühler et al. 2022, Sykes et al. 2022, Toft et al. 2024 etc., Statham & Campell 2025). On the other hand, avalanche trigger zone estimation
model showed to work well and to be accurate when used in avalanche frequency estimation, although its accuracy is directly reliant on the precision of trigger estimation results.

Simulated runouts are relevant only for "Tiny" avalanches, based on the RAMMS:Avalanche calibration tables (SLF 2024), as avalanches of bigger measures might overrun simulated run-outs. Although, simulated run-outs may already be too long for skier-triggered avalanches as RAMMS:Avalanche is not calibrated for small i.e. skier-triggered avalanches ($\leq$ size 3).
For more accurate simulations of skier-triggered avalanches RAMMS:Extended model should be utilized (Dreier et al. 2014). The focus of the simulations was predominantly on dry slab avalanches, without accounting for the run-out characteristics of wet avalanches. However, it is noteworthy that dry slab avalanches represent the most significant danger to freeride skiers and are the most lethal, as indicated by the SLF (2025). Finally, the starting points of the avalanches were selected randomly from the areas model estimated to be avalanche-prone, not specially selected by avalanche experts.

In the overall assessment, apart from avalanche danger, extreme terrain was taken into account. Nonetheless, certain critical elements such as terrain traps, the potential of remote triggering, and the presence of protruding rocks were not



incorporated into the assessment. The inclusion of these factors could enhance the precision of the evaluation (Harvey et al. 2018). Furthermore, while straightforward and easily interpretable, it falls short of providing detailed information regarding the shape, direction, length, and frequency of potential avalanche paths, rendering it somewhat oversimplified. Although the
assessment effectively conveys general information concerning the avalanche danger in freeride zones, a map portraying avalanche paths would offer a more comprehensive understanding of the potential avalanche scenarios within these areas.

## 6 Conclusions and outlooks

This study presents a comprehensive approach to assessing avalanche hazard in freeride skiing zones, specifically
within the Jasná ski resort region of the Low Tatras. By integrating a validated avalanche trigger estimation model with advanced RAMMS run-out simulation model and current high-resolution topographic and vegetation data, the research offers a detailed spatial analysis of avalanche-prone areas and their frequency of occurrence. The results demonstrate strong alignment with the existing avalanche cadastre while also identifying additional risk zones, particularly on southern slopes.

Looking forward, several improvements can enhance both the precision and applicability of the method. One key
limitation is the dependency on manually mapped vegetation data, which is time-consuming and may hinder large-scale implementation. This could be addressed by incorporating semi-automated vegetation classification techniques or by using elevation model subtraction methods (DSM - DEM) to estimate vegetation cover more efficiently. In case of run-out simulations, a better calibrated model for small avalanches, RAMMS:Extended, could be used. Additionally, while the simulations focused on dry slab avalanches, the approach currently does not account for larger or wet avalanches, which may
follow different trajectories. Future work could expand on this by incorporating simulations for varying avalanche types and conditions. Furthermore, including other aforementioned auxiliary factors could improve hazard estimation accuracy.

Lastly, an important future step would be to transition the simulation framework to an open-source environment. Replacing RAMMS with AvaFrame (Tonnel et al. 2023) would significantly enhance the accessibility and reproducibility of the methodology. This shift would allow broader use by researchers, mountain guides, and local authorities, particularly in
regions where access to proprietary software is limited.

In summary, while the presented approach provides a strong foundation for freeride zone hazard assessment, its future development lies in increasing automation, expanding terrain considerations, adapting to a wider range of regions, and embracing open-source tools like AvaFrame to ensure broader impact and accessibility.

*Data availability:* Input, output and reference data described in this paper are publicly available on Zenodo (https://doi.org/10.5281/zenodo.16961371, Kupec and Koco, 2025).

*Author contribution:* AK (conceptualization, data curation, formal analysis, investigation, methodology, validation, writing (original draft preparation)), ŠK (conceptualization, data curation, project administration, methodology, supervision, writing
(review and editing)).



*Competing interests:* The authors declare that they have no conflict of interest.

*Acknowledgements:* This research was supported by Grant Agency for Doctoral Students and Young Researchers of the
University of Prešov (grant no. GaPU 14/2025). We would also like to express our gratitude towards SCAP for allowing us to
use their avalanche records for this study.

*Financial support:* This research was supported by Grant Agency for Doctoral Students and Young Researchers of the
University of Prešov (grant no. GaPU 14/2025).

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
