# Peer review of "Assessment of avalanche hazard of freeride skiing areas"

_EGUsphere, 2025_

## Referee Comment (RC1)

**General comments:**

My understanding of the paper is that it wants to improve the classification of the 12 freeride zones surrounding the Jasná ski resort in Slovakia. The current classification uses the terms Easy, Moderate and High, defined from the Jasná Freeride manual from 2017. There is no description of this classification in the current paper. The objective of this study is to develop an avalanche terrain map specifically for skier triggered avalanches. They do this by using the avalanche release zone estimation model by Biskupič and Barka (2009) which is based on terrain factors such as forest density, slope incline and curvature, but also descriptive avalanche statistics from accidents in the area. The avalanche release zone estimation model incorporates the likelihood of triggering an avalanche by identifying elevation bands and aspects which are more frequent in the avalanche incident data. The implications of this are not discussed. The output is classified into low, medium, high and very high release probability. Medium, high and very high cells are considered a release area if there is a cohesive unit of cells (not defined). From these release areas, a RAMMS simulation is done with the aim of estimating size 3 avalanches. The avalanche paths within each of the 12 zones is then again compared with the same avalanche incident data to divide the 12 zones into four categories: (1) safe, (2) moderately dangerous, (3) dangerous and (4) very dangerous. There seems to be no explanation behind why these classes are selected, how they are defined, or whether any other terrain hazard systems like the Avalanche Terrain Exposure Scale was considered. The lack of clear definitions from the suggested end classification makes it difficult to see the scientific significance of the paper. Several key concepts are not described or discussed in a balanced way. Furthermore, there are several typos and poor formulations throughout the manuscript making it hard to understand all concepts presented.

I also have some conceptual problems with the methods used in this manuscript.

1. The altitude and aspect factors are clearly defined in Table 1, but it is unclear where these values come from. My understanding is that they are based on SCAP accident data. If so, using accident data directly assumes that skiers are evenly distributed across all elevations and aspects. For example, the altitude factor is highest between 1650 and 1850 meters, likely because most accidents occur there—but this could simply reflect that more skiers are present at those elevations. The same issue applies to aspect: perhaps more people ski on north-facing slopes to find dry powder. Without accounting for this, the analysis risks going into the trap of "base rate neglect", which should be addressed and discussed.

   One reduction factor within the RM suggests the avoidance of North-facing, due to the more frequent accidents. This would be valid if there was an evenly distributed background traffic, or base rate. If the base rate is unknown, one might argue that there is a tendency for backcountry enthusiasts to favor North-facing slopes due to the superior snow conditions (e.g., Grímsdóttir & McClung, 2006). Luckily for the RM, Winkler et al. (2021) found North-facing slopes to be two times riskier compared to South facing slopes when adjusting for base-rate. Alarmingly, fatality data from Norway contradicts the more frequent accidents in North-facing slopes, indicating a higher accident incidence on South-facing slopes. However, the base rate remains unknown (Aasen, 2023). This is an example which showcase that including this type of information into the potential release model is a bad idea.

2. The assumption that a higher probability of triggering directly leads to more frequent avalanche occurrences can be misleading. For example, unstable slopes in remote or rarely skied terrain may never be loaded or triggered and thus produce few avalanches. Similarly, explosive control often reveals high sensitivity in slopes that would not be released naturally. Persistent weak layers may allow easy triggering, but if traffic is low, only a few avalanches are reported. Spatial variability also means that only isolated hot spots are highly trigger-sensitive, limiting overall avalanche frequency. Finally, short periods of high instability may pass without natural triggers or skier presence, further weakening the link between triggering probability and observed avalanche frequency. This should be discussed.

3. You first build a potential release area model using data on observed avalanches to identify which release areas are most frequent versus less frequent. You then simulate avalanches from all these areas. Because the avalanche data show that north-facing slopes at higher elevations are most common, these are treated as the most dangerous. Finally, you use the same avalanche data again to validate the modeled avalanche paths. In other words, the descriptive values from the avalanche occurrence data are used both to define the release areas and to check the model results. Then it's not that surprising that you find a correlation, or is there something that I have missed?

Please see individual line comments for more information. I have provided some examples of poor language, but it's too many to identify all through line comments.

**Line comments**
Line 7: Define inbounds/on piste skiing. "Crowded ski slopes" could be both ski touring and on piste skiing.

Line 8-9: "… greater dangers" greater than what?

Line 10: Freeride or ski-touring? In line one you define both freeride skiing and ski touring. Use terminology consistently.

Line 12: You state that the «… potential run-out paths for skier trigger avalanches ($\leq$ size 3) were modelled» In Line 299 you state that RAMMS is not calibrated for avalanches smaller than size 3. How is this then valid?

Line 22: Example of poor language: "**In** recent years, a notable **in**crease **in in**terest **in** adrenaline sports has been observed". Rewrite, repetitive language.

Line 22-33: Citations?

Line 25: Example of poor language: "… some skiing enthusiasts find overcrowded slopes of ski resorts less appealing and choose to explore unmaintained terrains through freeride skiing".
Example: some skiers find piste/inbounds skiing less appealing because of crowded slopes and therefore prefer sidecountry/backcountry/off-piste/freeride terrain. Select and define your terminology and be consistent.

Line 27: What is conventional skiing?

Line 28: The "skiing enthusiasts" are now "participants" in what?

Line 28: … risk of avalanches

Line 30: Example of poor language: "Therefore, the mitigation of the hazard through localization of hazardous areas and creation of easily interpretable representations should be our main focus".
Example: Therefore, our main focus should be to reduce the hazard by identifying dangerous areas and creating maps that are easy to understand.

Line 45: According to Sykes et al. 2022, the two most state of the art PRA modelling methods are Bühler et al., 2018 and Veitinger et al., 2016. Both of these should be mentioned and you should argue why you consider the one you use is appropriate for your study, and how it works conceptually compared to the two others.

Sykes, J., Haegeli, P., & Bühler, Y. (2022). Automated snow avalanche release area delineation in data-sparse, remote, and forested regions. *Natural Hazards and Earth System Sciences*, *22*(10), 3247-3270.

Line 48-49: I would add D'Amboise et al. (2022) and Sampl and Zwinger (2004) as examples of avalanche runout model.

D'Amboise, C. J., Neuhauser, M., Teich, M., Huber, A., Kofler, A., Perzl, F., ... & Fischer, J. T. (2022). Flow-Py v1. 0: a customizable, open-source simulation tool to estimate runout and intensity of gravitational mass flows. *Geoscientific Model Development*, *15*(6), 2423-2439.

Sampl, P. and Zwinger, T.: Avalanche simulation with SAMOS, Ann. Glaciol., 38, 393–398, https://doi.org/10.3189/172756404781814780, 2004.

Line 53-55: Toft et al. (2024) uses the FlowPy runout model from D'Amboise et al. (2022). Toft et al. (2024) is cited, but the FlowPy model is not mentioned.
Lines 55–56: You cite Schmudlach and Köhler (2016) and Harvey et al. (2018) to support the claim that Bühler et al. (2022), Toft et al. (2024), and Statham and Campbell (2025) rarely focus on freeride or backcountry skiers. However, since the first papers were published before the latter ones, the argument as written does not make sense.

There are numerous papers published from 2018 until today's date that focus entirely on backcountry skiers (i.e., Thumlert and Haegeli, 2018; Larsen et al. 2020; Sykes et al., 2020; Hendrikx et al., 2022; Toft et al. 2024; Sykes et al. 2024; Harvey et al. 2024).

Thumlert, S., & Haegeli, P. (2018). Describing the severity of avalanche terrain numerically using the observed terrain selection practices of professional guides. *Natural hazards*, *91*(1), 89-115.

Toft, H. B., Sykes, J., Schauer, A., Hendrikx, J., & Hetland, A. (2024). AutoATES v2. 0: Automated Avalanche Terrain Exposure Scale mapping. *Natural Hazards and Earth System Sciences*, *24*(5), 1779-1793.

Sykes, J., Toft, H., Haegeli, P., & Statham, G. (2024). Automated Avalanche Terrain Exposure Scale (ATES) mapping–local validation and optimization in western Canada. *Natural Hazards and Earth System Sciences*, *24*(3), 947-971.

Harvey, S., Christen, M., Bühler, Y., Hänni, C., Boos, N., & Bernegger, B. (2024, September). Refined Swiss avalanche terrain mapping CAT v2/ATH v2. In *Proceedings International Snow Science Workshop, Tromsø, Norway* (pp. 23-25).

Line 57-58: What about avalanches larger than size 3? How are these dealt with?

Line 58-60: Move to methods section.

Line 58-60: Why do you use RAMMS compared to other simulation tools?

Line 58-60: Your approach using RAMMS to delineate size 3 avalanches is very similar to the one from Harvey et al. 2018; 2024. You should credit this.

Line 72: Reference to Figure 1 is missing in text.

Figure 1: The overview map (top right) should be separated from the rest of the information to the right (i.e., green on this map is not forest, but elevation lower than 100m?). North arrow and scale bar missing. The middle map is ok. The three lower maps should have a clearer north arrow and scale bar. I would also add a description that explains that these views are in 3D from a different direction compared to the main map above.

Line 82: Describe what the terms «easy, medium and difficult» actually means/how it's defined, and how it compares to ATES which is another common framework the categorize avalanche terrain.

Line 85: Reference to Figure 1 is here, the reference should be written before the figure is presented.

Line 87: How many incidents are there? How many fatalities?

Line 98: I assume the roughness input factor you are mentioning here is an input factor in RAMMS? And that the land cover (previously in text/figure termed forest) have to be made to successfully incorporate forest density/cover in RAMMS?

Line 95 vs Line 100: Define pixel size using the same notation.

Line 101-102: If avalanche records collected from the SCAP were crucial for calibration, you should add some more context regarding this input data here. How many avalanches? What data are associated with each avalanche? What is relevant for this study?

Line 103: I see no statistical analysis described in this section.
Line 106-109: I would like a figure that shows an example avalanche from this dataset, including the key information.

Line 110 (same as line 103): My understanding is that you have not done any statistical analysis on the SCAP data? Instead, the SCAP dataset provides relevant statistics on avalanche accidents. Is this correct? If any statistical analysis is done, please explain what you have done.

Section 3.3: I do not quite understand how this done without reading Biskupič and Barka (2009).

My understanding is that you have 6 input rasters, which are:
Al – Altitude factor
Ex – Exposition/Aspect factor
Fx – Profile curvature factor
Fy – Plan curvature factor
S – Slope inclination factor
RG – Roughness factor (proxy for forest density/vegetation cover/land cover)

The altitude and aspect factors are clearly defined in Table 1, but it is unclear where these values come from. Are they based on SCAP accident data? If so, using accident data directly assumes that skiers are evenly distributed across all elevations and aspects. For example, the altitude factor is highest between 1650 and 1850 meters, likely because most accidents occur there—but this could simply reflect that more skiers are present at those elevations. The same issue applies to aspect: perhaps more people ski on north-facing slopes to find dry powder. Without accounting for this, the analysis risks base rate neglect, which should be addressed and discussed.

Profile/plan curvature, slope inclination and roughness factor – how is this done? Which tools or code do I have to use to recreate these values before they are assigned according to Table 1?).

In the end, Av is defined as all areas larger than 15 cells, which would mean that the minimum release area is 15 square meters given the DEM input raster size?

Line 137-138: I think you have a conceptual problem with your "avalanche trigger zone estimation". I don't think you can call it a "probability of avalanche release" map because you do not know whether you have an even distribution of skiers at all elevations and aspects. See section 3.6 and 3.7 in Toft (2024) for more examples of base rate neglect.

Toft, H. B. (2024). Who skis where, when? UiT The Arctic University of Norway. https://hdl.handle.net/10037/35819

Line 143-144: I don't think you need this sentence: It serves as an alternative to conventional one-dimensional avalanche simulation tools such as AVAL-1D.

Line 147: What do you mean by skier triggered avalanches? Aren't all potential avalanche trigger zones from section 3.3 used for RAMMS simulations? If not, why not?

Line 160-161: The assumption that a higher probability of triggering directly leads to more frequent avalanche occurrences can be misleading. For example, unstable slopes in remote or rarely skied terrain may never be loaded or triggered and thus produce few avalanches. Similarly, explosive control often reveals high sensitivity in slopes that would not release naturally. Persistent weak layers may allow easy triggering, but if traffic is low, only a few avalanches are reported. Spatial variability also means that only isolated hot spots are highly trigger-sensitive, limiting overall avalanche frequency. Finally, short periods of high instability may pass without natural triggers or skier presence, further weakening the link between triggering probability and observed avalanche frequency. This should be discussed.

Line 161-168: Do you do a frequency estimation, or do you simply assign the frequency into four categories?

Line 170-172: How is this done, and why divide into four categories? Is this something new in this paper, or is it an existing scale?

Line 184: How do you deal with avalanches larger than size 3? Could terrain be classified as safe, while still being exposed to size 4 or 5 avalanches?

Line 182-183: Move to section 3.2?

Line 183-189: Move to discussion?

Line 190: I have still not seen a statistical analysis being described.

Figure 2: What is the difference between aspect of recorded avalanches and aspect (area)?

Line 201-206: What you describe is descriptive data, not a statistical analysis. You are reporting counts and percentages of avalanches in different elevation bands, along with measures of central tendency (mean and median). A statistical analysis would normally involve testing hypotheses, estimating confidence intervals, fitting models, or making inferences beyond the sample (e.g., regression, significance testing, survival analysis). So, in this case, it's more accurate to say that you present descriptive statistics (summary of the dataset) rather than a statistical analysis.

Line 201-206: Again, this is only observed avalanche activity within the study area. How does the percentage of terrain in your study area affect this? You will expect that if you have a very small percentage of your study area within 1250-1450 masl, you will also have a very small percentage of avalanches within this elevation band. How do you adjust for the background information? This should also be discussed in the discussion.

Line 208: What is the "estimation model"? Frequency estimation model, ref. section 3.5?

Line 208: So, if I understand correctly: you first build a potential release area model using SCAP data on observed avalanches to identify which release areas are most frequent versus less frequent. You then simulate avalanches from all these areas. Because the SCAP data show that north-facing slopes at higher elevations are most common, these are treated as the most dangerous. Finally, you use the same SCAP

data again to validate the modeled avalanche paths. In other words, the descriptive values from SCAP are used both to define the release areas and to check the model results. Then it's not that surprising that you find a correlation, or is there something that I have missed? Is the SCAP data something different from the "avalanche cadastre map"?

Line 255-274: In my view, this is the main result and not the beginning of the discussion.

Line 278: What is the trigger estimation? This is the model by Biskupič and Barka (2009)? You argue that this model is good because you only need a few inputs, but then you have to assume that Al, Ex and S factors would be relevant on all slopes all around the world (which I would argue they are not). That means that to use this model elsewhere, you would not only need a DEM and vegetation data, but also descriptive statistics on avalanche frequency (not going into the base rate neglect problem here).

Line 283: I would argue that the main limitation is the dependency of historical avalanche occurrence. You should also discuss this model compared to the one from Bühler et al., 2018 and Veitinger et al., 2016.

Line 297-299: You need to discuss how you deal with larger than size 3 avalanches, and why that is not a problem in your case.

328-330: Why did you not consider using AvaFrame for this study in the first place?

---

## Author Comment (AC2)

Dear Anonymous Referee #2,

Thank you for your review. We found your comments very constructive and believe that they will significantly strengthen the quality and clarity of our paper.

We have provided answers to your comments (your comments in bulleted italics) below:

• The topic—avalanche hazard assessment for freeride skiing areas—is relevant for mountain risk management. However, the scientific contribution and methodological soundness are questionable. The study mainly combines existing approaches (Biskupič & Barka model and RAMMS simulations) without introducing a clearly defined methodological innovation. Furthermore, the validity of the approach for freeride-scale avalanches is uncertain, given the limitations of the applied model and the arbitrary selection of simulation parameters.

We agree that our primary goal was not to develop a new model, but to clearly demonstrate and thoroughly measure how two established approaches can be used together in the specific context of *freeride* avalanche management. The combination of presented approaches was utilized in the past, however frequency estimation based on Biskupič and Barka model and RAMMS:Avalanche is new in our region. By full integration of aforementioned approaches we wanted to create a replicable framework for the objective and standardized assessment of local avalanche hazard at the scale of freeride terrain.

Regarding validity, we acknowledge the limitations of the RAMMS::Avalanche model when applied to small avalanches and therefore recommend the use of the more appropriate RAMMS::Extended model for future applications. We do not consider the random selection of release zones to be as problematic, as most of these release areas were identified only within regions marked as prone to release by the Biskupič and Barka model, and they also corresponded with the inventory produced by Milan, which was created based on expert assessment.

• Starting points for avalanche simulations were selected randomly rather than defined by expert judgment or objective terrain analysis. This introduces significant uncertainty and undermines reproducibility and physical realism.

We will substantially strengthen and refine the section on model limitations in the discussion.

After correction, we will explicitly state that we do not interpret the results of simulations as precise predictions, but rather as a pilot study (proof-of-concept) for spatial risk analysis, rather than for a calibrated prediction of avalanche runout.

• The RAMMS model version used (RAMMS:Avalanche) is not calibrated for small, skier-triggered avalanches (≤ size 3), which are the focus of this paper. The authors acknowledge this limitation but still base their conclusions on these simulations. No clear uncertainty or sensitivity assessment is provided. The validation (82.61 % overlap with cadastre) does not adequately measure model performance because both datasets may contain inherent spatial inaccuracies. Consequently, the results appear qualitative rather than quantitatively validated, and the methodology cannot be confidently generalized to other mountain areas.

We will emphasize that our study is intended as a pilot study (proof-of-concept) for spatial risk analysis, rather than a calibrated prediction of avalanche runout. The primary objective is to demonstrate and explore how existing modeling approaches can be applied in combination to assess relative spatial risk patterns. Consequently, our conclusions are based on relative, scenario-based outputs that illustrate potential hazard extents, rather than on precise predictive modeling of individual avalanche events. For

future work, which would utilize optimized model, we will recommend applying a more appropriate model, such as RAMMS::Extended, to improve accuracy of simulated run-outs.

We will completely remove the statement claiming 82.61% 'model performance validation.' Instead, in the Results section, we will refer to a 'spatial comparison with historical occurrences.' This comparison is intended solely to illustrate that the simulated runout zones correspond to areas where avalanches have historically occurred. In this case, the overestimation of run-outs by RAMMS: Avalanche does not impact the validity of the study, as it only illustrates the possibilities of integration of two models rather than a calibrated prediction of avalanche runout. Nevertheless, an uncertainty assessment of datasets, as well as, input data (DEM, vegetation data and avalanche records) will be added to the discussion.

We agree that the results are primarily qualitative/relative. This will be clarified in the Discussion. The methodology will explicitly state that the conclusions are based on relative spatial risk values intended for showcasing the possibility of model combination.

• The methods section provides many technical details but lacks a clearly structured, reproducible framework. Input data processing steps (DEM manipulation, vegetation classification) are described in detail, but the logical reasoning behind parameter choices is missing. The linkage between model inputs, assumptions, and outputs is weak. The approach's transferability to other freeride areas is not convincingly demonstrated.

We acknowledge the lack of reproductible framework, therefore, we will reorganize the Methods into a step-by-step workflow that explicitly links data inputs, parameter choices, modeling assumptions, and resulting outputs. We will add explicit reasoning for all key parameter selections (e.g., DEM preprocessing steps, vegetation classification thresholds, and release-zone criteria). The revised text now clearly explains how each input and assumption influences the model behavior and the resulting spatial risk patterns. We will add a subsection to the Methods that describes in detail the input criteria and GIS data-processing steps, which can be readily replicated in other mountain regions provided that DEMs, vegetation maps, and avalanche records are available.

• The discussion repeats descriptive results without deeper analysis or critical interpretation. It acknowledges model limitations but still presents the findings as reliable, which is inconsistent. The conclusion should explicitly state that the current model configuration has limited applicability to freeride conditions, rather than suggesting general adaptability.

In the Discussion, we will remove descriptive repetition and add a more critical evaluation of the results. We will clearly distinguish between what the simulations can reliably indicate (relative spatial patterns of potential hazard) and what they cannot provide (precise runout predictions for small, skier-triggered avalanches). We will emphasize that the outputs should be interpreted as scenario-based approximations rather than reliable event-level predictions. Lastly, the conclusion will be rewritten to explicitly state that the current model configuration has limited applicability to freeride conditions, unless RAMMS:Extended is used for accurate run-out modelling.